# Prediction of RF-EMF Exposure by Outdoor Drive Test Measurements

Shanshan Wang *[ID], Taghrid Mazloum [ID] and Joe Wiart [ID]

Chaire C2M, LTCI, Télécom Paris, Institut Polytechnique de Paris, 91120 Palaiseau, France;
taghrid.mazloum@telecom-paris.fr (T.M.); joe.wiart@telecom-paris.fr (J.W.)
* Correspondence: shanshan.wang@telecom-paris.fr

**Abstract:** In this paper, we exploit the artificial neural network (ANN) model for a spatial reconstruction of radio-frequency (RF) electromagnetic field (EMF) exposure in an outdoor urban environment. To this end, we have carried out a drive test measurement campaign covering a large part of Paris, along a route of approximately 65 Km. The electric (E) field strength has been recorded over a wide band ranging from 700 to 2700 MHz. From these measurement data, the E-field strength is extracted and computed for each frequency band of each telecommunication operator. First, the correlation between the E-fields at different frequency bands is computed and analyzed. The results show that a strong correlation of E-field levels is observed for bands belonging to the same operator. Then, we build ANN models with input data encompassing information related to distances to $N$ neighboring base stations (BS), receiver location and time variation. We consider two different models. The first one is a fully connected ANN model, where we take into account the $N$ nearest BSs ignoring the corresponding operator. The second one is a hybrid model, where we consider locally connected blocks with the $N$ nearest BSs for each operator, followed by fully connected layers. The results show that the hybrid model achieves better performance than the fully connected one. Among $N \in \{3, 5, 7\}$, we found out that with $N = 3$, the proposed hybrid model allows a good prediction of the exposure level while the maintaining acceptable complexity of the model.

**Keywords:** EMF exposure; AI; drive test; neural networks; RF

## 1. Introduction

The fast development of wireless communication technologies is accompanied by a rising concern about the risk perception of electromagnetic field (EMF) exposure induced by wireless network infrastructures. To respond to public concerns, many works [1–4] have been conducted to assess radio-frequency (RF) EMF exposure. Usually, this could be achieved by performing 'in situ' measurement campaigns [4,5], as well as exploiting data recorded by sensor networks. On the one hand, downlink (DL) and uplink (UL) EMF exposures are differentiated and monitored depending on the emitting source (i.e., base station antennas and mobile devices, respectively). On the other hand, RF-EMF exposure from different generations of network technologies such as the fourth generation (4G), small cells, and the fifth generation (5G) of wireless cellular networks are addressed.

In [4], the overall impact of the densification of macro cells with small cells on global human exposure (including both DL and UL) was studied by using two drive test solutions. In [6], RF-EMF exposure induced by RF base station (BS) antennas was assessed in an outdoor environment and an underground shopping mall in Japan. The measurement protocols are also being updated with the deployment of new 5G networks, as proposed in [7,8]. However, measurement campaigns were performed mainly for compliance at the designated locations with drawbacks, such as being time-consuming and expensive. For the spatial RF-EMF exposure in a complex environment, a global view is missing.

In addition to conventional measurement approaches, simulations and mathematical models were also well explored. In [9], simulation methods were adopted to assess RFEMF

exposure in urban public trams. In [10,11], surrogate models are proposed to assess exposure induced by 4G networks. In [12], stochastic geometry is used as a powerful mathematical tool to model the distribution of RF-EMF exposure strengths induced by 5G networks. The simulation and mathematical modeling tools clearly bring advantages in terms of cost reduction. However, the validity of models is questioned due to the simplification of environments or propagation models.

To overcome those limitations, new methods, such as artificial neural network (ANN) models, provide potential solutions from a data-driven point of view. Many machine learning or artificial intelligence (AI) related works have been dedicated to channel propagation, such as path loss prediction [13,14]. In those papers, distance from the receiver to the source is the key parameter. Moreover, other features are extracted from the physical environment and fed to AI algorithms to predict propagation channel characteristics and perform optimization [15,16]. Different from the prediction of propagation characteristics from single transmitters, EMF exposure deals with aggregated received powers from multiple BSs.

To the best of the authors' knowledge, the applications of AI in the field of RF-EMF exposure are still very new but are receiving growing interest. Very rare works have started to exploit AI to predict, on the one hand, the power emitted by a mobile phone (equivalent to UL exposure) [17,18] and, on the other hand, the DL exposure [19–21]. Regarding the UL exposure, the ANN model of [17] and the machine learning models of [18] are fed with easily available parameters such as the DL connection indicators, e.g., reference signal received power (RSRP), and other information related to the environment. They considered realistic data by carrying out indoor measurement campaigns and outdoor drive test measurements, respectively. In [19], machine learning is exploited for the estimation of DL and UL EMF exposure in multi-source indoor WiFi scenarios. In [20], the DL exposure map in indoor environments is reconstructed using convolutional networks. Both proposed models [19,20] were validated on data collected from simulations. Indeed, ANN-based predictions either require prior knowledge of measurements or are only valid using simulations. Predicting the DL RF-EMF exposure based on realistic measurement data and only requiring publicly accessible information is not well addressed.

In our previous work [21], we studied the possibility of utilizing an ANN model to predict exposure levels from simulated measurement data in an urban outdoor environment. In the present work, we intend to extend our work in [21] by conducting real-life drive test measurements in an outdoor urban environment. Broadband isotropic measurements of the E-field are performed over the frequency band from 700 to 2700 MHz in Paris. After data analysis, we performed RF-EMF spatial reconstruction by considering two different ANN models. The first one is a fully connected ANN model, where we take into account the $N$ nearest BSs ignoring the corresponding operator. The second one is a hybrid model consisting of locally connected blocks, where each block is dedicated to a given operator with $N$ nearest BSs as inputs. More common inputs consist of the position of the receiver and the time. Finally, the comparison between the two models, as well as the impact of the number of neighboring BSs ($N$) on the prediction accuracy, are addressed.

This paper is organized as follows: Section 2 explains the measurement system, setup, and drive test protocol used in the present paper. Section 3 introduces the structure of ANN models used to predict RF-EMF exposure. Section 4 gives the results of the analysis of drive test measurement data and the performance of ANN-based predictions. Section 6 concludes the paper.

## 2. Measurement Description

### 2.1. Measurement Equipment

Drive test measurement campaigns were carried out in Paris using a portable spectrum analyzer, i.e., RSA306B from Tektronix [22]. A wide frequency band from 700 to 2700 MHz was selected in the drive test, covering all RF bands from 2G, 3G and 4G networks. Isotropic measurements of the E-field are performed by connecting the one-port Tektronix to a 3-axis

dipole antenna via a switch. The 3-axis antenna was fixed on the top of the vehicle while conducting the drive measurement. The measurement equipment is shown in Figure 1.

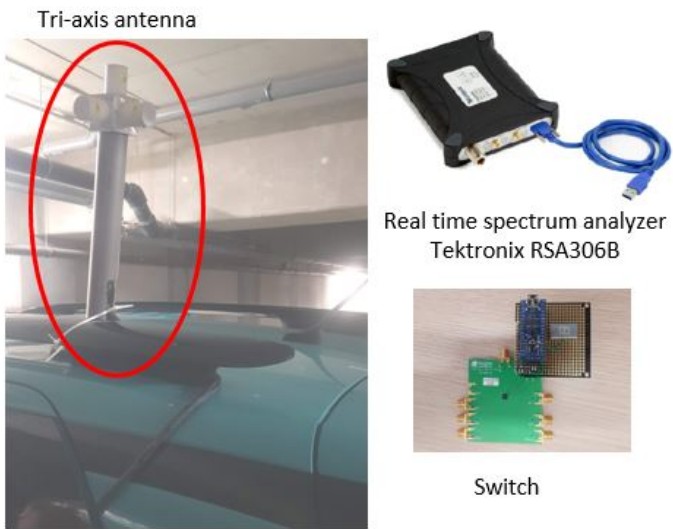

**Figure 1.** Measurement system: 3-axis antenna, Tektronix and switch.

The signal acquisition was performed on the Tektronix RSA306B, while the data analysis, storage and replay were performed on the laptop (PC) side. Managing the PC separately from the acquisition hardware makes processing upgrades easy and minimizes IT management issues. We developed a python-based graphical user interface (GUI) to configure and control the RF-EMF measurements by transferring the measurement setup to Tektronix. The setup information includes resolution bandwidth (RBW), reference level, center frequency and span. For each measurement record, the isotropic E-field strength is obtained by scanning over the three ports of the antenna and computed as $E = \sqrt{\sum E_j^2}$, where $j \in \{1, 2, 3\}$. Then, the process is repeated continuously. The measurement steps can be found in Figure 2.

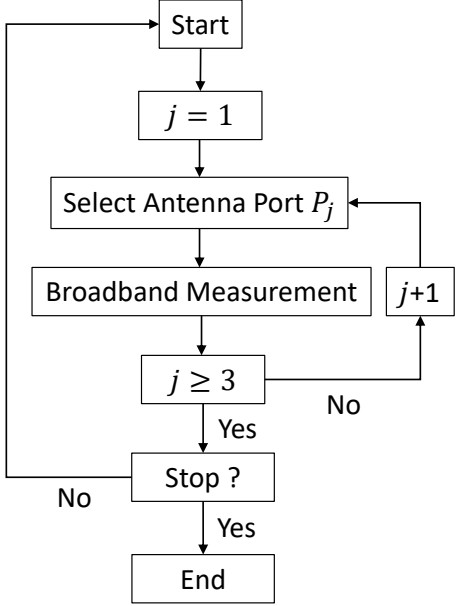

**Figure 2.** Flow chart of the measurement procedure.

## 2.2. Drive Test Protocol and Analysis

The drive test route (shown in Figure 3) is selected to increase the diversity of measurement environments in Paris. The total itinerary is approximately 65 Km in one day, from 9 a.m. to 5 p.m. The objective of the drive test is to conduct measurements covering a large area of Paris within a limited time. In other words, the challenge is to perform the measurements on the three antenna axes at almost the same position, which depends mainly on the driving speed. In our case, the average driving speed is 9 Km/h (i.e., 2.5 m/s), which allows for reducing the discontinuity of 3-axis measurements while bringing the least inconvenience possible to the road traffic. The average distance between two successive measurement points is around 3 or 4 m.

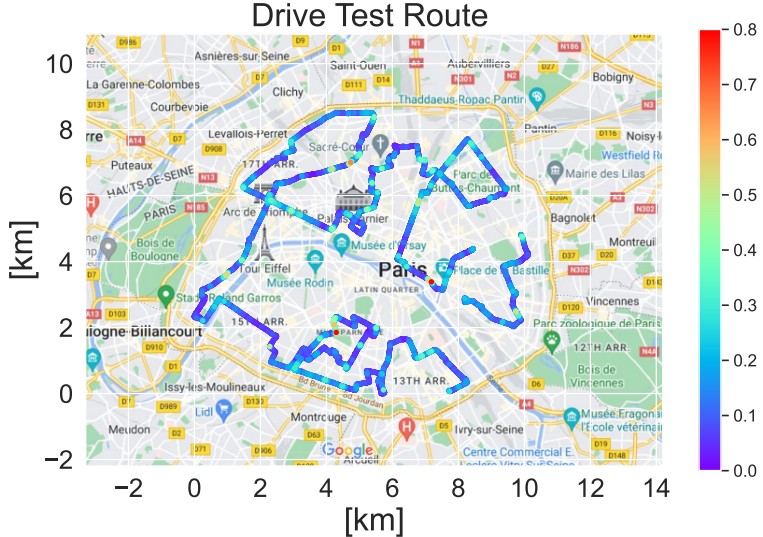

**Figure 3.** Drive test route in Paris.

The GPS information is recorded using a mobile phone application, i.e., 'Geo Tracker' since Tektronix does not have a built-in GPS module. Geo Tracker records Universal Time Coordinated (UTC) and coordinates (Longitude and Latitude) information, which are later used to be synchronized with Tektronix data. Then, the GPS coordinates of each measurement record are obtained by: (1) Interpolation of the GPS information recorded by Geo Tracker based on time synchronization, (2) conversion from longitude and latitude to Universal Transverse Mercator (UTM) coordinates.

After the broadband drive test measurement was obtained, the processing and analysis were carried out. First, the E-field levels from RF cellular bands of 4 main French operators were extracted and computed. In Figure 4, we compare the correlation of E-field levels from each band. A strong correlation is observed among bands from the same operator. This could result from a deployment strategy in that each operator tends to put new antennas on an existing site for the consideration of cost. For this reason, we consider two different ANN models for the RF-EMF spatial reconstruction, which is explained in Section 3.1.

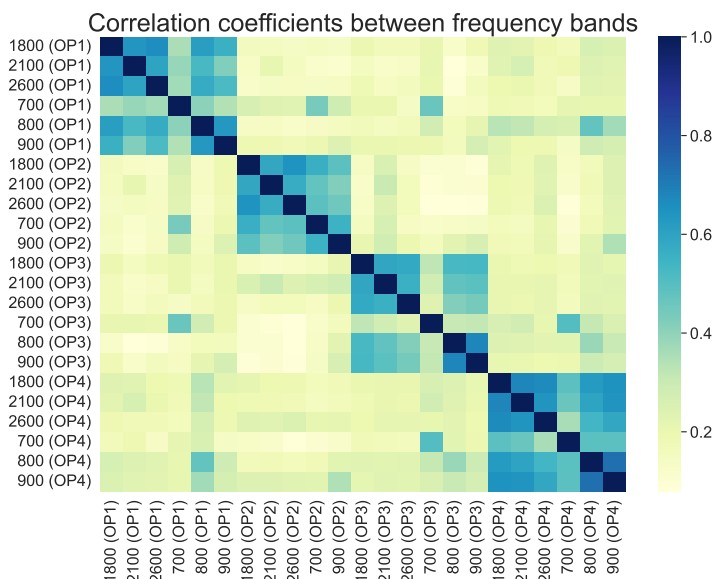

**Figure 4.** Heatmap of the correlation between bands of different operators.

According to guidelines given by ICNIRP [23], the reference levels for EMF exposure from 100 KHz to 300 GHz in the whole body case are frequency-dependent since they depend on the equivalent surface of the body, which is linked to the frequency. For example, reference limits at frequencues 400 MHz ($L_{4MHz}$) and 2 GHz ($L_{2GHz}$) are 27.5 and 61 V/m for the general public, respectively. As described in [23], the human body has a higher capability to grab and absorb EM energy in the frequency band (60–100 MHz). In this frequency band, the adult human body size (e.g., height between 1.5 and 1.8 m) is close to a quarter wavelength. Because of that, the admissible maximum power density must be lower than elsewhere. For a frequency above 2 GHz, the human absorption is much more local and less dependent on the frequency. In this case, the admissible maximum power density is constant. In this context, the root squared integration of the E-fields $E_1$ and $E_2$ measured, respectively, at the frequency bands $f_1$ and $f_2$ (given by $\sqrt{E_1^2 + E_2^2}$) cannot be used to check the compliance to the limit. Indeed, the objective is to verify that the sum of the percentages with respect to the relative reference limit is less than 100%. Considering the limits $L_1$ and $L_2$ at $f_1$ and $f_2$, the sources $S_1$ and $S_2$ are, respectively, inducing $(\frac{E_1}{L_1})^2$ and $(\frac{E_2}{L_2})^2$ percent of their corresponding limits. For frequency between 400 and 2000 MHz, limit $L$ is defined as $1.375\sqrt{f}$, where $f$ is the frequency in MHz. Therefore, we adopt the following metric to assess total exposure level [24]:

$$E_{total} = \sqrt{\sum_i^N \frac{f_{ref}}{f_i}(E_i)^2},\tag{1}$$

where $N$ is the number of frequency bands and $f_{ref}$ is the reference frequency of interest.

In the present paper, the E-field from each band is normalized to frequency $f_{ref} = 900$ MHz. Therefore, the total Equivalent 900 (Eq900) E-field is used to represent the overall RF-EMF exposure in the outdoor urban environment collected by drive test measurements.

After processing the data from drive test measurements, we can obtain the following features: (1) total E-field level in terms of Eq900, (2) time of each measurement point, (3) UTM coordinates of each measurement point. In addition to the features obtained via post-processing of measurements, the additional publicly accessible information about the BS is extracted from [25]. To summarize, with all the features and information obtained in this section, we plan to build ANN models and perform spatial reconstruction on RF-EMF exposure in an urban outdoor environment.

### 3. Spatial Reconstruction on EMF Exposure

In this section, we describe the ANN model selected for the spatial reconstruction of the EMF exposure. Then, we explain and discuss the different input parameters of the ANN model.

#### 3.1. ANN Model

We consider a feedforward ANN model, which consists of one input layer, several hidden layers, and one output layer. Each layer is composed of several artificial neurons, inside which an activation function is applied to the dot product between the output of the previous layer and weights $W_i$ (including bias). In the training process of the ANN model, the weights $W_i$ are optimized in order to minimize the loss function. However, one must notice that an ANN model cannot be overtrained to avoid overfitting.

The ground truth is the total Eq900, which takes into account all frequency bands from all operators. Accordingly, the ANN model's inputs include the relative distances from each measurement point to the $N$ nearest base stations. Additional inputs concern publicly accessible information, as mentioned in the previous section.

As mentioned in Section 2, whether taking into account the network operator or not when choosing the $N$ closest BSs motivates us to propose two different architectures for the ANN model. The first one is a fully connected ANN model, and the second one is a hybrid model with $M$ locally connected blocks followed by fully connected layers.

The two ANN models are illustrated in Figure 5 and described as follows:

**(a) Fully connected ANN model.** We consider the $N$ nearest distance to a mixture of all BSs without distinguishing operators, as the ground truth is used to reconstruct the total Eq900.

A fully connected feedforward neural network is used in this model (as seen on the left in Figure 5). Inputs $\{x_1, x_2, \ldots, x_N\}$ represent distances to $N$ nearest neighboring BSs, while $\{x_{P,1}, \ldots, x_{P,Q}\}$ represent hour of the measurement and normalized UTM coordinates.

**(b) Hybrid connected structure.** As seen in Figure 4, E-field strengths in different bands from each operator are highly correlated. It is of interest to utilize this information to help predict the total Eq900. Therefore, we sorted distance to neighboring BSs from each operator and adopted the same hybrid structure as proposed in [21] (as shown on the right in Figure 5). The hybrid design of locally connected (LC) blocks and fully connected (FC) blocks can reduce unnecessary interactions and enhance the interactions of correlated inputs. Input $x_{m,n}, m \in \{1, \ldots, M\}, n \in \{1, \ldots, N\}$ represents n-th nearest distance to neighboring BS from m-th operator. Then $\{x_{P,1}, \ldots, x_{P,Q}\}$ contain the same information as in the previous model. Here, the ground truth is the same as model (a), i.e., Eq900 computed from drive test measurements.

Furthermore, to explore the impact of $N$ on the prediction of the measured E-field strength, we consider different values for $N$ in the results.

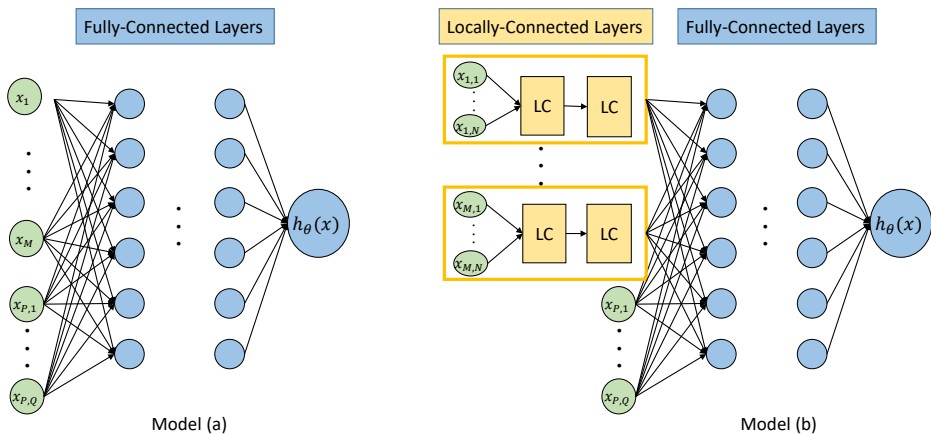

**Figure 5.** ANN structure for model (a) and model (b).

## 4. Results

In this section, the results of the spatial prediction of EMF exposure based on the ANN models described in Section 3 are presented. First, the analysis of data collected from drive test measurements is carried out, then the reconstruction performance based on the ANN models is studied for several scenarios.

### 4.1. Analysis of Drive Test Data

The drive test measurements along the route illustrated in Figure 3 recorded 19,834 points in total. The variation of the E-field strength along this route is shown on the left of Figure 6. The cumulative distribution functions (CDF) of Eq900 E-field from each operator ($E_{Opi}$) and total Eq900 ($E_{Total}$) are compared on the right of Figure 6. The Eq900 for each cellular operator ($E_{Opi}$) is calculated using Equation (1). We note here that the E-field represents the total Eq900, which is obtained from the integration of the E-field strength over all the cellular operators ($E_{Opi}$), computed as $E_{Total} = \sqrt{E_{Op1}^2 + E_{Op2}^2 + E_{Op3}^2 + E_{Op4}^2}$. We observe that Eq900 from each operator has almost similar distributions. Obviously, the total Eq900 presents higher values for the E-field strength because of the integration over all the operators. However, the CDF plots show that the RF-EMF exposure in the outdoor urban environment is well below the limit given by ICNIRP guidelines (over the frequency range 700 to 2700 MHz, the maximum limit is 61 V/m). The measurement point with higher values $E_{Opi}$ shown in an area tends to have close BSs of the corresponding operator. The measurement point with low values, on the other hand, tends to be far away from the BS or the antenna is directed elsewhere. Similarly, total Eq900 varies spatially according to the distribution and the density of the BSs in a given area. Moreover, the E-field strength strongly relies on the traffic.

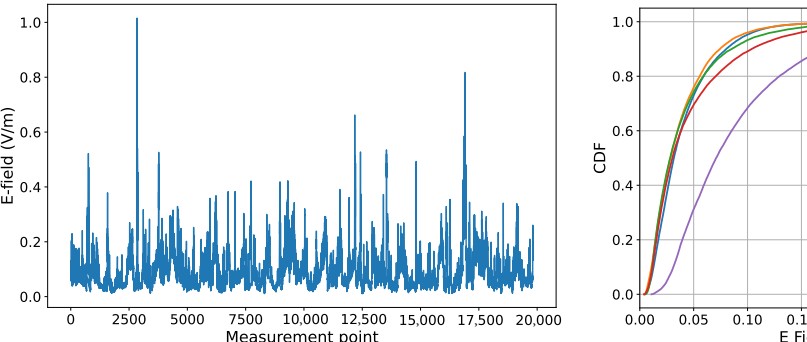

**Figure 6.** (**Left**): The variation of the total Eq900 along the drive path. (**Right**): CDF of total Eq900 and Eq900 from each operator.

In order to check the density of the BSs in Paris, we computed the mean distances between each measurement point and its *n*-th nearest neighboring BSs. The results given in Table 1 show a dense deployment of BSs in the urban city. This raises the question of how many neighboring BSs significantly influence the local exposure level in an urban environment. To answer this question, we study the impact of a number of neighboring BSs *N* on the ANN-based prediction performance of EMF exposure.

**Table 1.** Mean distance between each measurement point and neighboring BS.

| | 1st BS | 2nd BS | 3rd BS | 4th BS | 5th BS | 6th BS | 7th BS |
|---|---|---|---|---|---|---|---|
| Mean Distance (m) | 80.693 | 127.296 | 163.649 | 192.726 | 219.452 | 242.019 | 262.701 |

### 4.2. ANN-Based Prediction of RF-EMF Exposure

The proposed models (a) and (b) have, respectively, $N + 2$ and $4N + 2$ inputs. The number of outputs for both models (a) and (b) is one, e.g., total Eq900. Both ANN models were

tuned using the grid-search method with a 5-fold cross-validation. The hyper-parameters used for the two models in the current paper are shown in Table 2. The exponential decay learning rate is adopted with $k = 0.001$ when training the ANN model. The neural network is constructed with the Python interface Keras in the TensorFlow environment. The PC is equipped with GPU from NVIDIA Quadro RTX 8000. After the ANN model is trained, the model is then applied to the test data with 10-fold averaging to obtain performance metrics. The obtained metrics, i.e., mean square error (MSE) and $R^2$, are shown in Table 3. Here, $R^2$ measures how close the two distributions are. A higher $R^2$ indicates a better prediction performance.

**Table 2.** Hyper-parameters of the proposed ANN model.

| Hyper-Parameters | Model (a) | Model (b) |
|---|---|---|
| Number of layers | 5 | 3 (LC) + 3 (FC) |
| Number of neurons (Hidden layers) | 40 | 3N (LC) + 50 (FC) |
| Optimizer | Adamax | |
| Activation function | Relu | |
| Learning rate $l_{ini}$ | 0.01 | 0.005 |
| Learning Rate decay | $l = l_{ini} \exp^{-kEpoch}$ | |
| Number of epoch | 100 | |
| Train:Validation:Test | 0.49:0.21:0.3 | |
| Loss function | mean squared error | |

The results of both models are shown in Table 3. In the spatial reconstruction of total Eq900, we compared the prediction performance by considering different values of the $N$ nearest BSs with $N = \{3, 5, 7\}$. For model (a), there is a significant improvement comparing $N = 5$ to $N = 3$. While when $N = 7$, there is a slight improvement in prediction quality in terms of $R^2$. When $N$ increases, the complexity of the model and the data pre-processing time increase, as well as the time of the model training.

**Table 3.** Results of the prediction performance with different values of $N$.

| N | Model (a) | | Model (b) | |
|---|---|---|---|---|
| | $R^2$ | MSE | $R^2$ | MSE |
| 3 | 0.66522 | 0.00148 | 0.805 | 0.000805 |
| 5 | 0.74810 | 0.00111 | 0.806985 | 0.00079 |
| 7 | 0.75016 | 0.0011 | 0.813376 | 0.000764 |

In model (b), total Eq900 E-field is predicted by considering $N$ nearest BSs from each operator, as shown in Table 3. Increasing $N$ from three to seven yields a limited improvement in the prediction performance but increases the computation time and the model complexity. Consequently, $N = 3$ is the adequate choice for this model. Nonetheless, if we compare horizontally in Table 3, we observe a clear improved prediction accuracy under the same value of $N$. This could be explained by the higher correlation between E-fields from the same operator, as shown in Figure 4.

From Figures 7 and 8, we observe that predictions (figure on the right) from the ANN model can provide a good reconstruction performance with ground truth (figure on the left), which agrees with the metrics in Table 3. The proposed ANN model is able to predict the total Eq900 E-field level with MSE = 0.000805 and $R^2 = 0.805$ for $N = 3$. From the

comparison of the two figures, we can see the majority of the predicted E-field strength is low, which agrees with the CDF plot in Figure 6. There are a few measurement points with higher values that stand out, possibly due to the closer distance to neighboring BS. The proposed ANN model is able to reconstruct those higher values well.

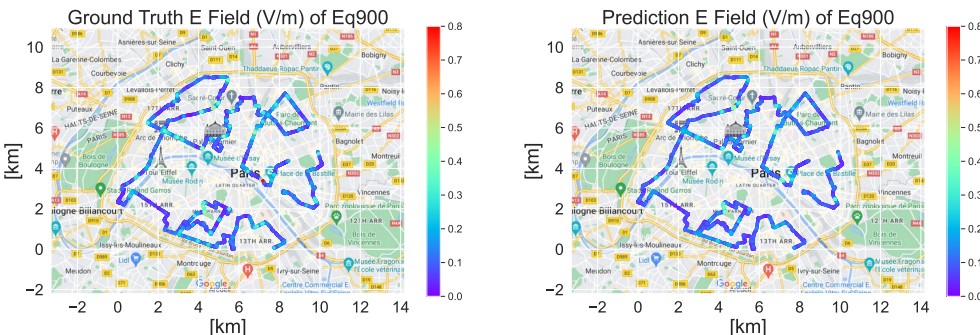

**Figure 7.** Ground truth (**left**) and prediction (**right**) of total Eq900 E-field prediction for $N = 3$ (and model b).

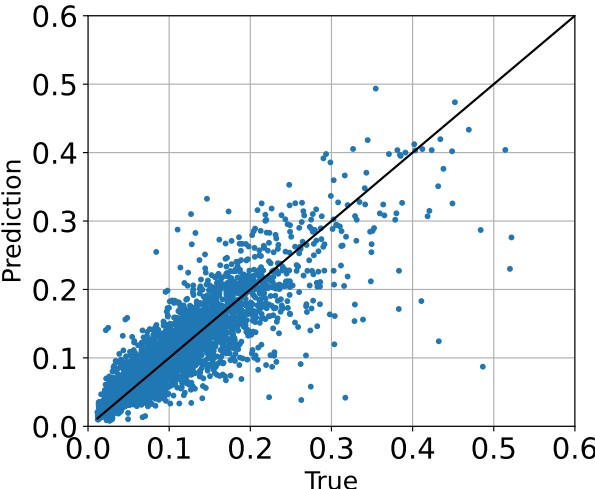

**Figure 8.** Scattering plot obtained from hybrid ANN with $N = 3$.

## 5. Discussion

The results show a good prediction in the outdoor urban environment using publicly accessible information. Indeed, considering the complexity of the ANN model and performance of prediction, the hybrid model with $N = 3$ is the best model for the reconstruction of RF-EMF exposure in terms of total Eq900.

However, the wireless traffic is time-varying [26,27], depending on the usage of voice and data. According to [26], the ratio of maximum to mean E-field at a given location and for a long duration is mainly below two. However, the time variation during the daytime is limited. As a consequence, the influence of the traffic on the drive test measurements along routes of 65 Km is limited compared to the influence of spatial location affected by the path loss. While in this paper we focus on the spatial reconstruction, we will dedicate future works to taking into account the temporal variation of the traffic.

Due to the limitation of devices for the current measurements, frequency selective drive test measurements with high resolution are not possible. To assess the RF-EMF exposure level in the frequency band of interest and improve the measurement design in the future, higher data acquisition resolution will be considered.

Since in the present paper we are interested in total Eq900, the impacts from channel fading are reduced in the computation of total Eq900. Therefore, large-scale fading domi-

nates the total Eq900, which indicates using distances to neighboring BSs as the input of the ANN models is reasonable. Moreover, the ANN model should consider more input parameters related to the surrounding environment, such as the building density, the average building heights, etc. Extensive measurement campaigns should be carried out in order to collect more data from various environments. Consequently, the model should be updated and tested on a small part of the route. All these aspects will be considered in future works.

## 6. Conclusions

In this work, we focus on the prediction of DL total RF-EMF exposure in the outdoor urban environment. A broadband (700 to 2700 MHz) drive test measurement was performed on the streets of Paris using a spectrum analyzer, i.e., Tektronix RSA306B, connected to a 3-axis dipole antenna via a switch. From the broadband measurements, E-fields for each frequency band of each operator were extracted and compared. The correlation between the E-fields indicates a strong correlation of E-field levels from bands belonging to the same operator. The measurement data were used to build an ANN model in order to predict the E-field strength. Eq900 was selected in order to account for the human body's capability of absorbing EM energy at different frequency bands and then to adequately represent the DL exposure. Then, two ANN models were built to predict total Eq900, considering $N$ nearest BS with and without distinguishing operators. The results show that increasing $N$ would increase complexity and computation time in training the ANN model while the performance of prediction is not always significantly improved. Moreover, we proved in the results that the hybrid model with $N = 3$ from model (b) is the best model for the current paper to use in assessing RF-EMF exposure levels in an outdoor urban environment.

**Author Contributions:** Conceptualization, S.W. and J.W.; data curation, T.M.; formal analysis, S.W.; investigation, S.W. and T.M.; methodology, S.W.; supervision, J.W.; validation, S.W.; writing—original draft, S.W.; writing—review and editing, S.W., T.M. and J.W. All authors have read and agreed to the published version of the manuscript.

**Funding:** The current work was partly supported by the following projects: Beyond 5G granted by Bpif and MINIRE.

**Institutional Review Board Statement:** Not applicable.

**Informed Consent Statement:** Not applicable.

**Conflicts of Interest:** The authors declare no conflict of interest.

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
