# Peer review of "Prediction of RF-EMF Exposure by Outdoor Drive Test Measurements"

_telecom, doi:10.3390/telecom3030021_

Round 1

Reviewer 1 Report

1. How does the neural network adjust the weightings? What is the algorithm? How many neurons are there in the input? The output? The hidden layer? Please explain in detail.

2. Since the neural network is used, what is the overall system architecture diagram? By using computer computing or embedded system computing?

3. The research results are too weak, please strengthen it.

4. The theoretical derivation is not rigorous enough, please strengthen it.

Author Response

Please check the attached pdf file for our replies.

Reviewer 2 Report

The paper deals with important problem - the artificial neural network (ANN) model for a spatial reconstruction of radio-frequency (RF) electromagnetic field (EMF) exposure in an outdoor urban environment. The results of this paper are valuable however paper needs improvement in terms of literature review. Very limited sources were cited and discussion section is very pour. The paper needs revision. Also some practical implications need to be provide in conclusions together with limits and future research guidelines to overcome these limits.

Author Response

(The authors gave the same response as above.)

Round 2

Reviewer 2 Report

The authors made somme corrections in their paper. Now manuscript is better quality and can be accepted in current form. The answers to my comments are provided and they satisfy my expectations.

Author Response

We thank the reviewer for the time spent on two rounds of review. We have further corrected some typos in the paper.